# Learning to Compose: Continual Visual QA through a Dual-Purpose Mixture-of-Experts Framework

## Abstract

Continual visual question answering with multimodal large language models is promising because of their strong reasoning and generative capabilities, but it remains hindered by catastrophic forgetting, concept drift across tasks, and the need for compositional generalization. Previous work has mainly targeted forgetting while overlooking the challenge of intertask composition, where real-world visual question answering requires combining knowledge across tasks. We introduce dual-purpose experts within a Mixture of Experts framework to address these challenges without the need for a replay buffer. Our approach expands expert layers in the multimodal space using low-rank adaptation and trains each expert jointly on Visual Question Answering and Visual Question Generation with a shared MLLM backbone. This unified design enriches multimodal knowledge, while knowledge sharing through the extraction and fusion of information from past experts further mitigates forgetting and enhances composition. A lightweight language-based router then enables effective expert selection. To better evaluate this setting, we also propose a compositional benchmark that reflects real compositional questions. Experiments on diverse benchmarks demonstrate that our method substantially reduces forgetting and improves compositional generalization compared to previous generative continual visual question answering approaches.

## 1 Introduction

Continual learning (CL) seeks to emulate human learning, where models acquire new knowledge sequentially while retaining prior information and avoiding catastrophic forgetting (CF), defined as the loss of previously learned knowledge when acquiring new tasks (French, 1999). Despite substantial progress, most continual learning research has focused on single-modal classification tasks with closed output spaces. In such settings, common strategies, including replay buffers , architectural constraints, or task-specific modules struggle to balance stability and plasticity (Wang et al., 2024).

Recently, multimodal architectures have gained traction for their ability to learn joint representations across modalities. Yet these models are also prone to forgetting, and most multimodal CL approaches remain confined to classification problems (Thengane et al., 2022). In parallel, the rise of multimodal large language models (MLLMs) has shifted attention toward reasoning and generative capabilities, offering a richer and more practical testbed for continual learning (Guo et al., 2025). Unlike closed-set classification, generative continual learning requires preserving both knowledge and reasoning skills while enabling transfer across tasks. This paradigm remains largely underexplored.

Visual Question Answering (VQA) exemplifies the challenges mentioned above. While VQA can be framed as classification, the generative formulation is more natural and scalable. To probe continual settings, Zhang et al. (2023) introduced a benchmark partitioning of VQA v2 (Goyal et al., 2017) into tasks defined by linguistically driven question types, thereby testing the retention of reasoning skills and multimodal understanding incrementally. Only a handful of works, notably VQACL (Zhang et al., 2023) and CL-MoE (Huai et al., 2025a), have explored generative continual VQA, with efforts focused primarily on mitigating forgetting. However, the open-ended nature of VQA presents challenges beyond retention: questions often require linguistic composition, naturally combining previously learned concepts. For example, a learner exposed first to a counting task

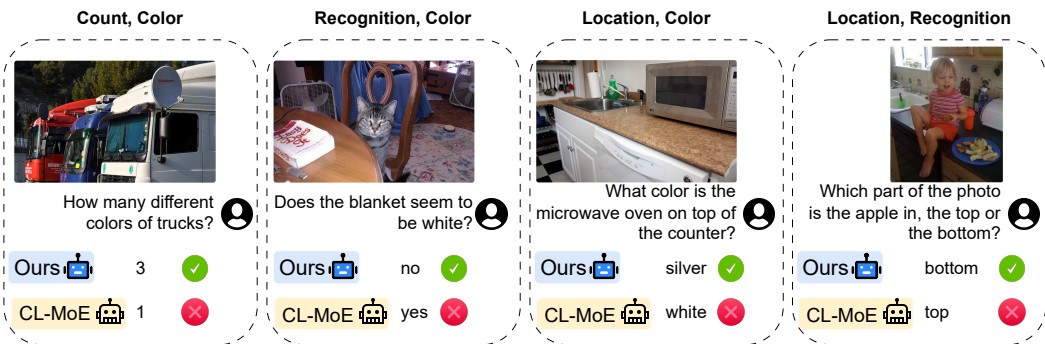

Figure 1: Comparison between Our MoE Design and CL-MoE on the Compositional Benchmark. For instance, in the leftmost column, when a user poses a query that necessitates the integration of skills from two distinct learned past tasks (Count and Color), CL-MoE fails. This limitation arises because the CL-MoE approach assumes that test instances originate from separate clusters; however, in real-world compositional settings, this assumption does not hold. In contrast, our design explicitly facilitates the understanding of task compositions, thereby enabling the model to generate more accurate and contextually appropriate responses.

(e.g., "How many trucks are there?") and later to a color task (e.g., "What color is the flip flop?") should be able to generalize to queries such as "How many different colors of trucks are there?" This reflects real-world scenarios, where tasks overlap and compose naturally during inference (soft task boundaries), rather than being predefined and isolated (hard task boundaries). Addressing such queries demands both compositional reasoning across seen tasks and richer visual understanding. Yet the VQACL benchmark lacks compositional Q&As (Hudson & Manning, 2019), leading prior work to overlook such scenarios and focus solely on per-task performance retention. For instance, while CL-MoE (Huai et al., 2025a) leverages mixtures of experts, its routing assigns queries to a single cluster and combines task- and instance-level weights using fixed parameters. This approach effectively addresses forgetting (i.e., drops below zero; see Table 1) but leads to brittle integration across multiple skills. As shown in Figure 1, even the current state-of-the-art method fails on simple compositions, underscoring a critical gap in current approaches.

Inspired by the success of the Mixture of Experts (MoE) framework in addressing forgetting, we adopt an MoE-based architecture and pose two central questions: (1) how can we maximize and preserve per-task performance, and (2) how can the model be designed to reflect compositional ability? To address the first, we introduce parameter isolation between experts, which stabilizes past knowledge and prevents forgetting. However, isolation reduces compositional ability, and a naïve alternative, parameter sharing across tasks, quickly becomes a bottleneck as the number of tasks increases, leading to overly general representations and renewed forgetting. Instead, we maintain isolation while enabling inter-expert communication by making experts dual-purpose, where visual question generation serves as an auxiliary task to visual question answering. As tasks grow, this dual role fosters teacher–student interactions that promote knowledge fusion while preserving isolated parameter spaces. Our unified architecture requires no memory buffer, reduces parameter overhead through low-rank adaptation, and achieves scalability and task-agnostic generalization with a lightweight language-based router.

To assess compositional reasoning, we created a human-annotated benchmark that combines two or more tasks into a single QA on the same test set images as the traditional VQACL benchmark, complemented by an additional public benchmark. Our method demonstrates effectiveness across three different benchmarks, maintaining task-specific knowledge, mitigating forgetting, and enabling compositional skills. Our contributions are as follows:

- **Unified dual-purpose MoE framework:** We propose an MoE-based architecture with parameter isolation to stabilize past knowledge and mitigate forgetting, while introducing dual-purpose experts with inter-expert communication to enhance compositional ability.
- **Efficient design:** Our approach avoids reliance on memory buffers, reduces parameter overhead via low-rank adaptation, and employs a lightweight language-based router to support task-agnostic generalization.

- **New compositional benchmark and empirical validation:** We introduce a human-annotated benchmark to assess compositional reasoning, and demonstrate that our method preserves task-specific knowledge, alleviates forgetting, and enables compositional skills across three different benchmarks.

## 2 RELATED WORK

Multimodal learning involves training models on two or more data modalities, such as audio, video, text, or images (Wu et al., 2023). When these models are built on top of large language models (LLMs) (e.g., OPT (Zhang et al., 2022), FlanT5 (Chung et al., 2024)) and process multiple modalities, they are referred to as multimodal large language models (MLLMs) (Alayrac et al., 2022; Bai et al., 2023). Examples such as BLIP-2 (Li et al., 2023), VLT5 (Cho et al., 2021), and PaLI-Gemma (Beyer et al., 2024) illustrate how models can generalize across diverse vision-language tasks. More recently, research has focused on enhancing the generative and reasoning capabilities of MLLMs, with GPT-4 serving as a prominent example (Achiam et al., 2023). Such generalization is particularly important in the continual learning (CL) setting, where models must learn a sequence of tasks without revisiting previously seen data (Wang et al., 2024). A core challenge in this setting is catastrophic forgetting, where models tend to lose prior knowledge while learning new knowledge (Kirkpatrick et al., 2017). Although MLLMs demonstrate impressive reasoning and generalization, they are not immune to the forgetting issue (Li et al., 2023). Classical continual learning strategies, including parameter regularization (Kirkpatrick et al., 2017), replay-based methods (Chaudhry et al., 2019; Buzzega et al., 2020; Zhang et al., 2023), and architectural adaptations (Rusu et al., 2016) have been proposed to address those issues, but they are not specifically tailored for MLLMs. This is a significant gap, as MLLMs are critical for tackling problems that require reasoning skills.

Moreover, existing continual learning benchmarks were not built for multimodal LLMs and do not fully capture their abilities. A practical and representative setting that requires reasoning skills is Visual Question Answering (VQA) (Antol et al., 2015; Whitehead et al., 2021), which assesses understanding both visual and textual modalities to produce open-ended, reasoning-driven responses. Zhang et al. (2023) introduced a continual VQA for generative answers, offering a more suitable evaluation setting for MLLMs. The authors proposed a memory-based method to tackle the challenges of continual VQA. However, this approach lacks scalability, and does not fully prevent catastrophic forgetting. Das et al. (2025) proposed a memory-free method based on generative questions. Nevertheless, the effectiveness of their approach is highly dependent on the quality of the generated questions and is further limited by catastrophic forgetting.

To mitigate forgetting, parameter isolation methods such as Mixture of Experts (MoEs) (Shazeer et al., 2017; Huai et al., 2025b) have shown promise. Recently, Huai et al. (2025b) tackled this using Dual-Router MoE (RMoE) and Dynamic Momentum MoE (MMoE). However, due to the limited number of shared parameters, they still experience performance degradation over time. At the same time, scaling up MLLMs has been shown to enhance performance (Li et al., 2023). Yet, their large parameter counts make retraining from scratch or incorporating new knowledge through multiple experts computationally impractical.

An alternative line of research, Visual Question Generation (VQG) (Li et al., 2018) leverages the generative reasoning capabilities of MLLMs to produce meaningful questions. Inspired by this, we remove the need for external memory entirely by fully exploiting MLLMs' generative capacity to preserve and transfer knowledge. Specifically, our approach focuses on information maximization through dual-purpose task experts, aiming to improve both reasoning and compositional skills while mitigating forgetting, without relying on memory.

## 3 PRELIMINARIES

We cast visual question answering (VQA) as a generative question answering task in a continual-learning (CL) setting. Unlike conventional offline training regimes, real-world systems must handle tasks that arrive sequentially, adapting to each without retraining from scratch. This online formulation is particularly challenging because the model must integrate new skills while retaining prior ones, without explicit access to past data. Formally, let $\mathcal{F}_{\boldsymbol{\theta}} : \mathcal{X} \to \mathcal{A}$ denote a pre-trained multimodal large language model (MLLM) with parameters $\boldsymbol{\theta}$ that maps an input space $\mathcal{X}$ of image–question pairs to a free-form textual answer space $\mathcal{A}$. The model encounters $T$ tasks (e.g., recognition,

counting), arriving one at a time. At step $t$, it is given only the current task $\mathcal{T}^t$, defined by the dataset $\mathcal{D}^t = \left\{ \left( (\mathbf{v}_i^t, q_i^t), a_i^t \right) \right\}_{i=1}^{|\mathcal{D}^t|}$, where $(\mathbf{v}_i^t, q_i^t)$ is an image–question pair and $a_i^t = (a_{i,1}^t, \ldots, a_{i,|a_i^t|}^t)$ is its tokenized answer sequence. The objective is to adapt $\mathcal{F}_{\boldsymbol{\theta}}$ to each new $\mathcal{T}^t$ without catastrophic forgetting of previous tasks, despite training only on $\mathcal{D}^t$ at each step. The model is finetuned sequentially on each task with the language modeling loss $\mathcal{L}$:

$$\mathcal{L}(\boldsymbol{\theta}) \;=\; \frac{1}{|\mathcal{D}^t|} \sum_{i=1}^{|\mathcal{D}^t|} \left[ -\frac{1}{|a_i^t|} \sum_{k=1}^{|a_i^t|} \log P_{\boldsymbol{\theta}}\big(a_{i,k}^t \,\big|\, a_{i,<k}^t, \mathbf{v}_i^t, q_i^t\big) \right], \quad t = 1, \ldots, T. \tag{1}$$

where $a_{i,<k}^t$ denotes the prefix tokens of the answer up to position $k-1$. Importantly, during evaluation, the task identity (e.g., recognition, counting) is *unknown*, making the continual setting more challenging than task-aware formulations.

## 4 METHODOLOGY

Our method, illustrated in Figure 2, unifies two objectives in a Mixture of Experts (MoE) framework. We introduce a parameter isolation technique with a knowledge fusion strategy for compositional reasoning, and define a language-based router for task-agnostic expert selection.

### 4.1 PARAMETER ISOLATION THROUGH MoEs

Inspired by the success of low-rank adaptation (LoRA) (Hu et al., 2022) in LLMs, we extend it to the multimodal space of MLLMs to construct task-specific experts within an MoE framework. Unlike prior approaches that combine parameter isolation and parameter sharing, the latter often reintroduces forgetting due to a shared parameter bottleneck. In contrast, our design leverages LoRA adapters to enforce complete isolation, meaning that experts remain independent during training even though new experts may be initialized from previous ones. This ensures that each expert preserves task-specific knowledge while avoiding the forgetting issue.

Concretely, we insert LoRA (Hu et al., 2022) adapters into the pretrained BERT-based Q-Former (Li et al., 2023) with a dual-level design, applying them to both self-attention and feed-forward layers. In self-attention, we augment the query ($\mathbf{W}_q$), key ($\mathbf{W}_k$), and value ($\mathbf{W}v$) projections to capture task-specific queries, which are passed through frozen cross-attention to extract visual features. Feed-forward adapters then translate these outputs into **task-specific** representations. Together, these modules form a single expert $\mathcal{E}_{\text{VQA}}^t(\cdot, \cdot)$ with parameters $\boldsymbol{\Phi}^t$. To train each expert, we use the language modeling loss from Equation 1, which can be reformulated as:

$$\mathcal{L}_{\text{VQA}}^t(\boldsymbol{\Phi}^t) \;=\; \sum_{i=1}^{|\mathcal{D}^t|} \left[ -\frac{1}{|a_i^t|} \sum_{k=1}^{|a_i^t|} \log P_{\boldsymbol{\theta}+\Delta\boldsymbol{\theta}(\boldsymbol{\Phi}^t)}\big(a_{i,k}^t \,\big|\, a_{i,<k}^t, \mathbf{v}_i^t, q_i^t\big) \right], \quad t = 1, \ldots, T. \tag{2}$$

Where $\Delta\boldsymbol{\theta}(\boldsymbol{\Phi}^t)$ denotes the task $\mathcal{T}^t$ specific low-rank updates applied across the designated projection and feed-forward layers. This design preserves knowledge because when a new task $\mathcal{T}^{t+1}$ arrives parameters $\boldsymbol{\Phi}^{t+1}$ starts from prior knowledge of $\boldsymbol{\Phi}^t$ while only optimizing $\boldsymbol{\Phi}^{t+1}$. As a result, the task-specific parameters of $\mathcal{T}^t$ remain frozen while allowing the new expert to adapt. However, as others remain frozen, there is no communication between experts, causing experts to learn in isolation. This isolation prevents forgetting at the cost of sacrificing compositional ability. In joint training, all task samples are available within a batch, enabling the model to build shared representations and implicitly acquire some compositional skills. In contrast, continual learning lacks access to prior task samples, and the multi-expert design further restricts implicit sharing because prior knowledge of $\boldsymbol{\Phi}^{t+1}$ is being overwritten either partially or completely by the current task. To overcome this, we build expert communication while maintaining expertise, effectively enabling independent experts with fusion, but without parameter sharing.

### 4.2 INTER-EXPERT KNOWLEDGE FUSION

We adopt knowledge fusion via teacher–student learning to bridge communication across experts. A naive solution is replay, but multimodal replay is memory-intensive since both visual and textual

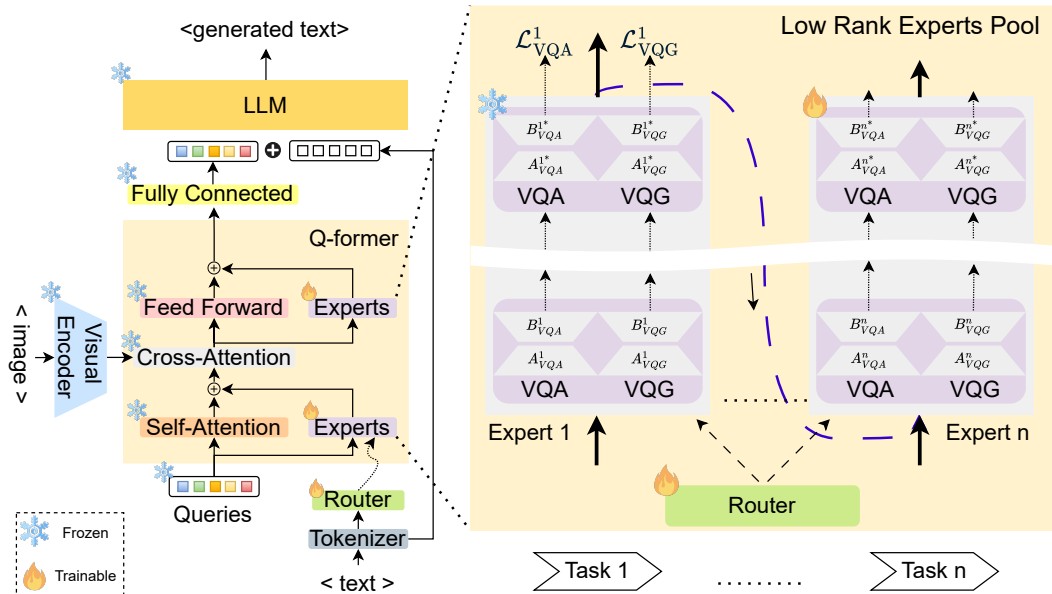

Figure 2: Overview of the proposed dual-purpose MoE framework. For both VQA and VQG, low-rank adapters are applied in parallel to the self-attention and feedforward layers. In the cross-attention layer, the experts' outputs, combined with the frozen self-attention output, serve as the query for the cross-attention mechanism, helping to extract task-specific features from the input image. Collectively, we refer to this as an Expert. During training, we optimize task loss $\mathcal{L}_{VQA}$ and $\mathcal{L}_{VQG}$, along with the LSTM-based router, while all other components except the task experts remain frozen.

data must be stored, and controversial due to privacy concerns. Instead, our knowledge fusion via teacher–student learning, where the expert (or an auxiliary generator) synthesizes pseudo-samples from prior tasks. In this dual role of generator and solver, the backbone enables access to past experiences without requiring raw data.

Prior studies suggest that teacher–student learning can transfer knowledge effectively without significant information loss (Robins, 1995; Shin et al., 2017). However, fine-tuning the low rank expert $\mathcal{E}_{VQA}^{t}(\cdot, \cdot)$ on the current VQA task induces catastrophic forgetting of question generation. Although the frozen backbone can still be utilized, its zero-shot distribution diverges from the task-specific distribution, leading to expert mismatch. Moreover, zero-shot generalization remains unreliable in low-resource or specialized domains (e.g., medical, legal, or scientific applications) due to limited domain coverage during pretraining. Following Li et al. (2018), who unified VQA and VQG, we introduce an auxiliary objective that aligns generated questions with the task-specific distribution, thereby mitigating distributional drift and improving continual knowledge fusion.

### 4.2.1 THE VQG EXPERT

To build a unified framework, we adopt the same architecture for the VQG module as for the VQA module. We extend our expert definition to incorporate both components, denoting each expert as $E^{t} = \{\mathcal{E}_{VQA}^{t}, \mathcal{E}_{VQG}^{t}\}$, $\psi^{t} = \{\mathbf{\Phi}_{VQA}^{t}, \mathbf{\Phi}_{VQG}^{t}\}$, where $\mathcal{E}_{VQA}^{t}$ and $\mathcal{E}_{VQG}^{t}$ are the task-$\mathcal{T}^{t}$ dual purpose components, respectively, and $\mathbf{\Phi}_{VQA}^{t}$, $\mathbf{\Phi}_{VQG}^{t}$ are their corresponding low-rank parameters. The VQG component is trained together using the same task dataset as the VQA component, but with input–output roles reversed. Specifically, instead of receiving a question and predicting an answer, the expert takes a text prompt and an image as input and generates the corresponding question. This setup poses no issue during training, since both question–answer pairs are available. However, to ensure that generated questions follow the real, task-specific distribution, we explicitly condition the model with a structured prompt of the form $p^{t} = $ `Generate a` $\{\mathcal{T}^{t}\}$ `question for the image where the possible answers are` $\{a^{t}\}$. The ground-truth questions from the dataset are used as the output target, and the expert is optimized using language modeling loss defined in Equation 3.

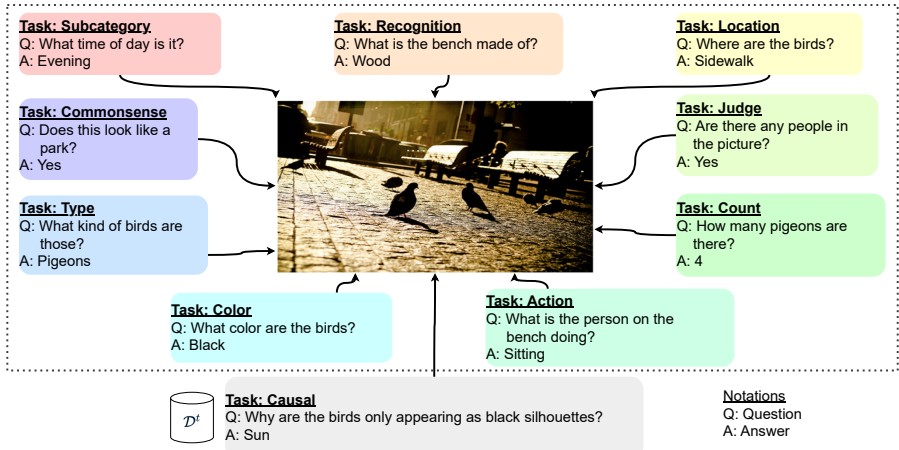

Figure 3: Compositional learning process. Traditional VQA is limited to a single-step learning setting (gray outer box), where each training instance is only a $\left((\mathbf{v}_i^t, q_i^t), a_i^t\right) \in \mathcal{D}^t$ triple. In contrast, our approach expands learning into a compositional space (dotted box), where multiple diverse Q&A types are asked about the same image. This enables richer multimodal reasoning, including **compositional skills**, leading to deeper understanding beyond the traditional scope.

$$\mathcal{L}_{VQG}^t = \sum_{i=1}^{|\mathcal{D}^t|} \left[ -\frac{1}{|q_i^t|} \sum_{k=1}^{|q_i^t|} \log P_{\boldsymbol{\theta}+\Delta\boldsymbol{\theta}\left(\boldsymbol{\Phi}_{VQG}^t\right)}\left(q_{i,k}^t \mid q_{i,<k}^t, \mathbf{v}_i^t, p^t\right) \right], \tag{3}$$

Here, $q_i^t = (q_{i,1}^t, \ldots, q_{i,|q_i^t|}^t)$ denotes the tokenized question sequence. Together equations 2 and 3 ensure expert $E^t$'s task-specific knowledge is preserved. During the continual learning process, $E^m$ used on $\mathbf{v}^t \in \mathcal{D}^t$ to generate $(\tilde{q}^m, \tilde{a}^m)$ where $m$ range from 1 to $t-1$. To ensure the quality of the generated questions, we employ confidence-based filtering. For each generated question $\tilde{q}_i^m = (\tilde{q}_{i,1}^m, \ldots, \tilde{q}_{i,|\tilde{q}_i^m|}^m)$, we define the confidence score as the average log-likelihood of its tokens:

$$C(\tilde{q}_i^m) = \frac{1}{|\tilde{q}_i^m|} \sum_{k=1}^{|\tilde{q}_i^m|} \log P_{\boldsymbol{\theta}, \boldsymbol{\Phi}_{VQG}^m}\left(\tilde{q}_{i,k}^m \mid \tilde{q}_{i,<k}^m, \mathbf{v}_i^t, p^m\right). \tag{4}$$

We retain only those generated questions whose confidence exceeds a threshold $\tau$, i.e., $\mathcal{Q}_{\text{filtered}}^m = \left\{ \tilde{q}_i^m \mid C(\tilde{q}_i^m) \geq \tau \right\}$. This filtering step discards low-confidence generations and ensures that only high-quality, task-relevant questions are used. Finally, for each question in $\mathcal{Q}_{\text{filtered}}^m$, we associate it with its corresponding image and leverage corresponding $\mathcal{E}_{VQA}^m$ to generate answers, resulting in question–image–answer triples that fuse the knowledge derived from past experts into the current task. We denote this as $\left\{ \left((\mathbf{v}_i^t, \tilde{q}_i^m), \tilde{a}_i^m\right) \mid \tilde{q}_i^m \in \mathcal{Q}_{\text{filtered}}^m \right\}$. Differently from existing methods, the fusion serves as a knowledge bridge rather than a knowledge-preserving replay. While parameter isolation preserves prior expertise, our approach enables independent experts to acquire task-specific question distributions that guide compositional reasoning. As shown in Figure 3, each expert generates complementary views of an image from a shared backbone expert pool, providing supervision that strengthens multimodal reasoning and facilitates compositional knowledge fusion across tasks.

Overall, our dual-purpose expert within a unified architecture addresses both catastrophic forgetting and compositional reasoning. To further enable task-agnostic generalization, we introduce a language-based routing mechanism that dynamically assigns inputs to the appropriate experts.

### 4.3 LANGUAGE-BASED ROUTING

To enable task-agnostic inference, we introduce a *language-based router*. The intuition is that the surface form of a question contains strong semantic cues about the underlying task (e.g., counting, recognition, color), which can be exploited to predict which expert should be activated without relying on explicit task identifiers at inference time. Given a tokenized question sequence $q$ of length $|q|$, we embed the tokens through a learnable embedding layer and process them with a recurrent encoder. Specifically, a single-layer LSTM encodes the sequence into a hidden representation $h_q \in \mathbb{R}^{d_h}$,

which is passed through a fully connected layer to produce task logits: $r(q) = \mathrm{softmax}(W_{\mathrm{fc}}\, h_q)$, where $W_{\mathrm{fc}}$ is the classification head. The predicted task is given by $\hat{t} = \arg\max r(q)$, which is then used to select the corresponding expert $E^{\hat{t}}$. Because the number of tasks grows over time, the router is trained in an incremental fashion with corresponding $E^t$. The training objective is standard cross-entropy over the expert pool (i.e., seen tasks) as targets:

$$\mathcal{L}_{\mathrm{router}} = -\frac{1}{|\mathcal{Q}^t|} \sum_{i=1}^{|\mathcal{Q}^t|} \log r(q_i^t)[t], \tag{5}$$

where $\mathcal{Q}^t$ denotes the set of training questions at task $\mathcal{T}^t$, and $t$ is the ground-truth expert index for question $q_i^t$. The router itself is also susceptible to forgetting. To retain past routing capabilities in the continual setting, we also ensure teacher-student knowledge fusion from $\mathcal{E}_{\mathrm{VQG}}^{\leq t}$ experts (See Section: 3) to the router. Specifically, for each past task $\mathcal{T}^m$, we include $\tilde{q}^m$, ensuring that the router preserves discriminative ability across all previously seen tasks. This integration makes the router robust to forgetting, while maintaining negligible overhead since training is performed solely on a single modality. Consequently, the router provides a scalable and task-agnostic mechanism for expert selection during inference. Figure 2 shows the overview of our proposed framework.

# 5 EXPERIMENTS

We first evaluate our method on the standard VQACL setting. We then extend the evaluation to the linguistic compositional setting and perform comprehensive ablation studies.

## 5.1 EXPERIMENTAL SETUP

**Backbone & Baselines & Metrics.** In our experiments, we adopt MLLMs as backbones. Following prior studies, we evaluate three representative architectures: VL-T5 (Cho et al., 2021), LLaVA-7B (Liu et al., 2023), and BLIP-2 (Li et al., 2023). We benchmark performance under standard continual learning baselines. For replay-based methods, we consider ER (Chaudhry et al., 2019), DER (Buzzega et al., 2020), and VS (Wan et al., 2022), while for regularization-based methods, we include EWC (Kirkpatrick et al., 2017) and MAS (Aljundi et al., 2018). In addition, we compare against frameworks designed specifically for open-ended language generation, including VQACL (Zhang et al., 2023), GaB (Clustering) (Das et al., 2025), and the most recent CL-MoE (Huai et al., 2025a). We include Vanilla, which refers to incrementally training without any continual learning method. Joint Training means training all tasks together. In our case, the **upper bound** is when, for a sample, any expert generates the correct answer. We adopt two standard continual learning metrics: Final Average Performance (AP) and Average Forgetting (AF) (Zhang et al., 2023). Let $\alpha_{i,j}$ denote the test performance on task with index $j$ after completing training on task with index $i$, and $T$ the total number of tasks. AP measures overall performance after continual fine-tuning, defined as $AP = \frac{1}{T}\sum_{t=1}^{T}\alpha_{T,t}$. AF quantifies knowledge loss on past tasks, defined as $AF = \frac{1}{T-1}\sum_{t=1}^{T-1}(\alpha_{t,t} - \alpha_{T,t})$.

**Implementation Details.** We use BLIP-2 (Li et al., 2023) as the backbone MLLM architecture, with ViT-g/14 (Fang et al., 2023) as the frozen vision encoder and FlanT5-XL (Chung et al., 2024) as the large language model. The multimodal Q-Former block of the backbone is configured with a fixed query size of 32. For our MoE implementation, we apply LoRA (Hu et al., 2022) for low-rank adaptation, fine-tuning only the LoRA layers on the self-attention and feed-forward modules of the BLIP-2 Q-Former block. We set the LoRA rank to 64 with a dropout rate of 0.1. During evaluation, we use the prompt template *"Question: {} Short answer: {}"*. The VQG threshold $\tau$ is set to the 90th percentile. For memory-based methods, to ensure fair comparison, we fix the replay buffer size to 5000 across all baselines. For the MoE-based baseline CL-MoE (Huai et al., 2025a), to ensure fairness, we reproduce the results with the official codebase by setting the number of experts to 8, while keeping all other settings the same as reported in the original paper. We use publicly available results whenever possible. See Appendix D for more on implementation details.

## 5.2 MAIN RESULTS ON THE TRADITIONAL VQACL

**Setting.** We conduct our experiments in the VQACL setting (Zhang et al., 2023), which is derived from the VQA v2 benchmark (Goyal et al., 2017). The VQA v2 dataset consists of open-ended,

Table 1: Performance comparison of different methods and backbones on the VQACL tasks. We report both average performance (AP) and average forgetting (AF) in percentage (%).

| Methods | Rec. | Loc. | Jud. | Com. | Cou. | Act. | Col. | Typ. | Sub. | Cau. | AP ($\uparrow$) | AF ($\downarrow$) |
|---|---|---|---|---|---|---|---|---|---|---|---|---|
| VL-T5 backbone | | | | | | | | | | | | |
| Vanilla | 7.39 | 4.94 | 22.29 | 32.30 | 0.71 | 12.14 | 12.10 | 10.69 | 27.29 | 15.10 | 14.49 | 30.15 |
| EWC | 6.73 | 8.43 | 27.22 | 47.10 | 0.14 | 12.40 | 1.76 | 10.98 | 31.05 | 11.85 | 15.77 | 28.38 |
| MAS | 30.81 | 8.07 | 25.50 | 4.00 | 31.90 | 32.39 | 26.24 | 24.75 | 19.85 | 2.75 | 20.56 | 21.97 |
| ER | 18.64 | 21.36 | 61.27 | 64.17 | 30.29 | 52.84 | 43.39 | 23.31 | 42.75 | 11.85 | 36.99 | 4.80 |
| DER | 14.55 | 13.83 | 62.88 | 65.16 | 30.96 | 51.19 | 40.51 | 19.04 | 42.87 | 12.55 | 35.35 | 6.58 |
| VS | 15.66 | 19.21 | 59.86 | 32.16 | 27.28 | 47.79 | 32.32 | 20.44 | 41.38 | 10.20 | 34.03 | 11.68 |
| VQACL | 22.83 | 19.82 | 30.66 | 44.24 | 28.25 | 42.82 | 64.13 | 66.5 | 57.2 | 15.2 | 39.16 | 2.42 |
| LLaVA-7B backbone | | | | | | | | | | | | |
| Vanilla | 19.25 | 14.81 | 54.59 | 56.97 | 24.23 | 46.20 | 27.58 | 26.09 | 36.47 | 18.89 | 32.51 | 20.69 |
| EWC | 28.12 | 23.02 | 61.50 | 61.08 | 26.13 | 54.29 | 23.65 | 32.25 | 44.97 | 17.83 | 37.28 | 15.27 |
| MAS | 31.54 | 22.09 | 60.85 | 46.32 | 32.48 | 56.47 | 30.05 | 35.69 | 42.73 | 18.83 | 37.71 | 14.91 |
| ER | 29.31 | 25.74 | 63.46 | 65.78 | 31.92 | 58.39 | 45.17 | 34.55 | 46.24 | 18.96 | 41.95 | 10.20 |
| DER | 26.95 | 21.43 | 64.88 | 66.17 | 31.01 | 55.92 | 44.60 | 32.85 | 47.09 | 20.74 | 41.16 | 11.28 |
| VS | 28.48 | 24.09 | 61.37 | 67.20 | 29.56 | 54.64 | 33.98 | 32.91 | 45.82 | 19.89 | 39.79 | 12.70 |
| VQACL | 34.14 | 32.19 | 66.15 | 63.00 | 33.01 | 60.91 | 34.64 | 38.48 | 47.94 | 24.42 | 43.49 | 9.10 |
| CL–MoE | 34.54 | 32.34 | 67.83 | 65.28 | 34.70 | 61.40 | 34.31 | 40.88 | 49.56 | 22.58 | 44.34 | -0.01 |
| BLIP-2 backbone | | | | | | | | | | | | |
| ER | 45.76 | 25.64 | 72.87 | 66.91 | 15.35 | 67.12 | 53.28 | 50.70 | 49.87 | 15.20 | 46.36 | 9.75 |
| DER | 44.87 | 24.06 | 71.12 | 65.41 | 38.78 | 60.52 | 46.17 | 36.77 | 48.90 | 14.90 | 45.15 | 8.93 |
| GaB | 28.03 | 29.05 | 79.99 | 77.8 | 45.29 | 65.3 | 59.96 | 32.3 | 50.02 | 11.6 | 47.93 | 2.40 |
| VQACL | 47.92 | 25.70 | 75.95 | 69.86 | 41.42 | 64.63 | 49.31 | 39.27 | 52.22 | 15.92 | 48.22 | 5.76 |
| Ours | 47.78 | 35.02 | 79.43 | 74.18 | 48.23 | 73.20 | 69.53 | 56.38 | 57.06 | 20.0 | 56.08 | -0.43 |
| Joint Training | 48.33 | 32.73 | 78.89 | 73.82 | 45.41 | 71.96 | 64.69 | 54.45 | 56.00 | 17.50 | 54.37 | - |
| Upper Bound | 58.42 | 47.72 | 90.28 | 88.02 | 61.58 | 83.36 | 77.48 | 64.08 | 64.09 | 31.34 | 66.64 | - |

human-annotated question–answer pairs grounded in images, comprising approximately 1.1M questions over 200k images sourced from the MS COCO dataset (Lin et al., 2014). Within VQACL, the benchmark is partitioned into ten linguistically driven ten tasks: *Recognition*, *Location*, *Judge*, *Commonsense*, *Count*, *Action*, *Color*, *Type*, *Subcategory*, and *Causal*. This division ensures a hard task boundary. See Appendix C for more details about the benchmark.

**Analysis.** We report the main results on the traditional VQACL setting in Table 1. The benchmark defines task boundaries through linguistic cues and evaluates continuous reasoning adaptability across diverse tasks. It also presents varying levels of difficulty. Among them, causal reasoning is the most challenging and produces the lowest AP. Our results show this varying difficulty trend, as performance varies across task types and some categories remain consistently difficult. We organize the results by backbone. VL-T5 has the smallest parameter count, while LLaVA-7 billion and BLIP-2 (5 billion) are comparable in scale, although most of their parameters remain frozen. Our method achieves the **highest AP** of **56.08%** ($\Delta$ +16.29% increase compared to second best) and the **lowest forgetting** among all methods, while remaining competitive or superior on individual tasks. These results show that parameter isolation effectively **preserves past knowledge and reduces forgetting**. The negative sign indicates that, on average, knowledge gain surpasses forgetting. This phenomenon arises in the MoE-based method under open-ended settings (such as generative VQA), where any expert can produce the correct answer, as illustrated in the Upper Bound row.

## 5.3 RESULTS ON THE COMPOSITIONAL VQA FOR CL

**Setting.** While VQACL provides fine-grained linguistic tasks, it does not explicitly target compositionality (Hudson & Manning, 2019). In VQACL, tasks in the train, validation, and test splits are divided by hard linguistic boundaries. For example, the question "How many people are there?" is classified only under the `count` task. In contrast, we focus on composition, where a single question may require knowledge from multiple past tasks. For instance, if a CL model is trained on `recognition` and `location` tasks, we evaluate whether it can answer questions that require knowledge from both (e.g., "Where is the bottle that is filled with water?"). This setting better reflects real-world scenarios, as humans naturally compose knowledge rather than strictly querying about a single task. We measure compositional performance using two benchmarks. The first is the GQA dataset (Hudson & Manning, 2019), which contains compositional questions over real-world images. Unlike VQACL,

Table 2: Impact of the VQG component on task performance, where Improvement denotes gains with VQG over without, and Self Improvement reflects intra-task performance.

| Component | Rec. | Loc. | Jud. | Com. | Cou. | Act. | Col. | Typ. | Sub. | Cau. |
|---|---|---|---|---|---|---|---|---|---|---|
| w/o VQG | 41.25 | 29.49 | 74.72 | 69.27 | 3.17 | 66.16 | 52.31 | 45.91 | 48.63 | 19.82 |
| w/ VQG | 43.62 | 31.05 | 77.67 | 72.53 | 36.14 | 71.27 | 54.92 | 51.11 | 51.31 | 20.28 |
| Improvement | +2.37 | +1.56 | +2.95 | +3.26 | +32.97 | +5.11 | +2.61 | +5.2 | +2.68 | +0.46 |
| Self Improvement | - | +3.99 | +1.19 | +1.06 | +1.21 | +1.66 | +0.57 | +0.99 | +1.93 | +0.46 |

which is constructed from `human-annotated` Q&As, GQA relies on structured scene graphs and an `automated question engine`. To complement this, we construct a `human-annotated` subset of compositional Q&As (referred to as compositional VQA) from MS COCO images, providing a more natural evaluation setting. Additional details for both benchmarks are provided in the Appendix C.

**Analysis.** In Figure 4, we compare our method with three other MLLM-based generative continual VQA approaches on two compositional benchmarks. Our method achieves the best performance on both COCO-GQA and human-annotated compositional VQA. Specifically, our method improves over prior methods by a large margin on COCO-GQA, reaching 63.78% compared to 56.68% for the next best method. On the human-annotated compositional VQA benchmark, our method also leads with 54.76%, showing consistent gains. The CL-MoE lags on COCO-GQA than compositional VQA, likely due to distributional change

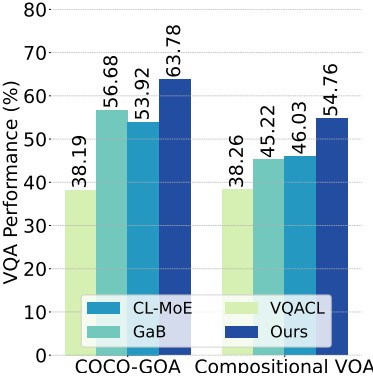

Figure 4: Compositional performance.

between the task questions and benchmark questions. Overall, figure 4 demonstrates the effectiveness of our inter-expert knowledge fusion.

## 5.4 ABLATION STUDY

**Impact of VQG.** One of the key components of our method that bridges communication through teacher–student knowledge fusion is VQG. Consequently, the quality of knowledge being fused directly impacts overall performance. To illustrate its effect, we evaluate the last expert's per-task performance. As shown in Table 2, incorporating VQG consistently enhances outcomes across all tasks. We also report the self-improvement in the last column, which reflects within-task performance gains and offers supporting evidence for Figure 3. See Appendix E for more details.

We evaluate the efficiency and scalability of our model in terms of the number of trainable parameters per task, as reported in Table 3. In our approach, three distinct components are fine-tuned when a new task arrives This design requires updating only 39.17M parameters, which represents a significant reduction compared to the 53.03M

Table 3: Task trainable parameters.

| Method | #Trainable Params ↓ |
|---|---|
| CL-MoE | 53.03M |
| Ours | 39.17M |

parameters required by alternative MoE-based framework. This corresponds to an approximate 26% decrease in parameter burden during task training, highlighting the efficiency of our method.

## 6 CONCLUSION

In this paper, we introduce dual-purpose experts within the Mixture of Experts (MoE) framework for continual visual question answering. Our proposed parameter isolation with a knowledge transfer mechanism mitigates the forgetting effect and enhances compositional generalization. The unified design avoids reliance on external memory or replay buffers. Finally, the routing mechanism ensures task-agnostic inference. We also propose a human-annotated compositional benchmark to evaluate the method on real-world images. Through extensive experiments on both compositional and standard VQACL benchmarks, we show that our approach consistently matches or surpasses baselines. However, compositionality in the current framework remains limited to past task knowledge within the continual learning setting. Still, the results highlight a promising path toward unseen compositional generalization.

ETHICS STATEMENT

This study relies on publicly available datasets and does not include human subjects or personally identifiable information. We follow the ICLR Code of Ethics, and the authors report no conflicts of interest.

REPRODUCIBILITY STATEMENT

We provide implementation details in the main paper, with hyperparameters and training procedures described in the appendix. Our code is linked in the supplementary materials to facilitate reproducibility.

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

## A  APPENDIX

We organize the appendix into several sections to provide comprehensive supporting material. First, we provide additional details of the expert architecture. Then, we present details on each benchmark: compositional VQA, the COCO GQA benchmark, and the traditional VQACL. Next, we provide additional implementation details, including workstation setups to aid reproducibility. We also include additional experiments, such as task dependency studies. Finally, we discuss the limitations of our method and provide instructions to access the source code.

## B  ADDITIONAL DETAILS OF EXPERT ARCHITECTURE

We define each expert as $E^t = \{\mathcal{E}^t_{VQA}, \mathcal{E}^t_{VQG}\}$, $\quad \psi^t = \{\mathbf{\Phi}^t_{VQA}, \mathbf{\Phi}^t_{VQG}\}$, where $\mathcal{E}^t_{VQA}$ and $\mathcal{E}^t_{VQG}$ are the task-specific dual-purpose components, respectively, and $\mathbf{\Phi}^t_{VQA}$, $\mathbf{\Phi}^t_{VQG}$ are the purpose specific corresponding low-rank parameters. However, each component shares the same underlying architecture. Specifically, we adopt LoRA (Hu et al., 2022) as the low-rank adapter.

As described in Section 4.1, we apply an adapter in the multi-modal space of MLLMs. Now denote a given layer with input $x \in \mathbb{R}^{d_{in}}$ and weight matrix $W \in \mathbb{R}^{d_{out} \times d_{in}}$, the original transformation is $y = Wx$. With LoRA, an adapter introduces a low-rank update $\Delta y = A(Bx)$, where $A \in \mathbb{R}^{d_{out} \times r}$ and $B \in \mathbb{R}^{r \times d_{in}}$ are low-rank matrices ($r \ll \min(d_{in}, d_{out})$). The final output is computed as $y' = Wx + \alpha \cdot A(Bx)$, where $\alpha$ is a scaling factor. In our expert setting, each task $\mathcal{T}^t$ is assigned an isolated low-rank parameter space while the pre-trained backbone $\theta$ remains frozen (i.e., both vision encoder and LLM). A component (i.e., VQA or VQG) output for task $\mathcal{T}^t$ is thus parameterized as:

$$\mathcal{E}^t(\mathbf{v}^t, q^t) = \mathcal{F}_{\theta + \Delta\theta(\mathbf{\Phi}^t)}(\mathbf{v}^t, q^t),$$

where $\Delta\theta(\mathbf{\Phi}^t)$ denotes the collection of all low-rank updates applied across the designated projection and feed-forward layers.

## C  ADDITIONAL DETAILS OF BENCHMARK

### C.1  VQACL BENCHMARK

The VQACL benchmark is based on the VQA v2 dataset (Antol et al., 2015). The VQA v2 contains human-annotated question and answer pairs over real-world images. The continual visual question answering (i.e., VQACL (Zhang et al., 2023)) benchmark is based on the VQA v2 benchmark's Karpathy splits of MSCOCO Karpathy & Fei-Fei (2015); Lin et al. (2014). Zhang et al. (2023) specifically designed ten linguistically driven tasks: *Recognition*, *Location*, *Judge*, *Commonsense*, *Count*, *Action*, *Color*, *Type*, *Subcategory*, and *Causal*. This design creates language-based hard task boundaries. However, images are shared between multiple tasks. The splits contain 131,478 (*Recognition*), 12,580 (*Location*), 160,179 (*Judge*), 25,211 (*Commonsense*), 62,156 (*Count*), 33,633 (*Action*), 50,872 (*Color*), 23,932 (*Type*), 31,594 (*Subcategory*), and 5,868 (*Causal*) train set questions. See Table 4 for dataset statistics.

### C.2  COCO GQA BENCHMARK

The GQA (Hudson & Manning, 2019) dataset is based on compositional question answering. The compositional setup requires multi-step reasoning to answer a question. The dataset, however, is created using the Visual Genome scene graph and question engine. The Visual Genome shares many images with the COCO dataset, which was used in the VQACL benchmark. The VQACL test task questions are based on 5,000 COCO images. Based on those images, we filter GQA questions. A total of 41,981 questions from the GQA test set share the same images as the VQACL test tasks. We select all of these questions as the COCO-GQA compositional benchmark. See Table 4 for dataset statistics.

### C.3  COMPOSITIONAL VQA

**Background.** Existing benchmarks have notable limitations for compositional reasoning in VQA. For example, the VQACL (Zhang et al., 2023) benchmark does not include compositional questions,

Table 4: Statistics of Traditional Linguistic-Driven VQA Tasks and Compositional Linguistic-Driven Tasks (all tasks share the same image source: MS COCO).

| Type | Task | Test Set | Examples |
|---|---|---|---|
| Linguistic | Recognition | 5628 | What is the chair made of?, What is on the floor? |
| | Location | 611 | Where is the giraffe?, Where is the food served in? |
| | Judge | 7194 | Is the boy playing baseball?, Are the windows big? |
| | Commonsense | 1100 | Could this be a multi-purpose room?, Has this pizza been baked yet? |
| | Count | 2658 | How many pictures are there?, How many chairs are in the photo? |
| | Action | 1373 | Is the man looking at the camera?, Is he happy or sad? |
| | Color | 2192 | What color are the gym shoes?, What color is the flip flop? |
| | Type | 1089 | What kind of room is this?, What kind of flowers are here? |
| | Subcategory | 1416 | Name the type of flower that is in the vase?, What animal is on the man's shirt? |
| | Causal | 200 | Why do the boats not have their sails up?, Why are all these zebra together? |
| Compositional | COCO GQA | 41981 | What is the person in front of the walls doing? (Location, Action), Is the bird brown and small? (Color and Judge) |
| | Compositional VQA | 200 | Does the person with red cap holding a camera? (Judge, Action, Color), How many people are there not carrying a bag in hands? (Count, Action) |

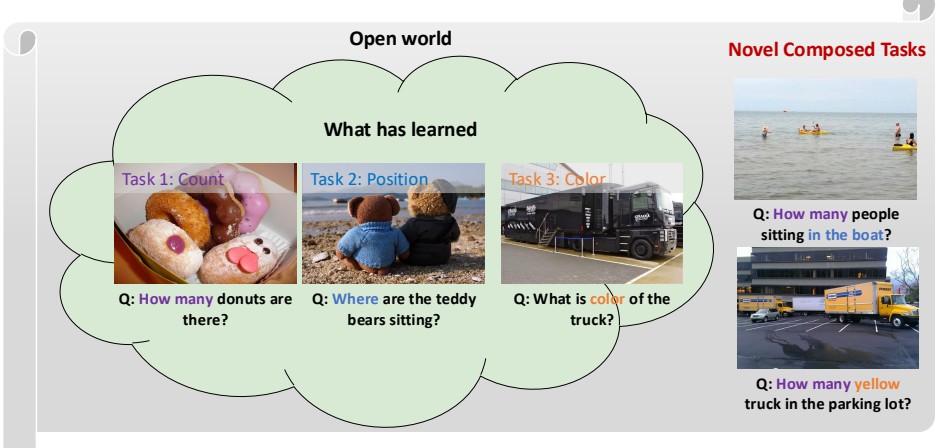

Figure 5: **Compositional Visual Question Answering (VQA) Challenge.** The model is trained incrementally on isolated tasks such as counting, object position, and color recognition. At test time, it must generalize to novel, composed queries that require combining previously learned concepts (e.g., "How many yellow trucks are in the parking lot?"). This illustrates the core challenge of continual VQA: robust compositional reasoning under continual task exposure without revisiting past data.

while GQA (Hudson & Manning, 2019) is automatically generated (not human-annotated) and was not designed with continual learning in mind. Consequently, these datasets lack explicit tags identifying compositional tasks (i.e., task identities are not available). In contrast, compositional questions are more practical and reflective of real-world settings. In practice, a VQA learner acquires tasks sequentially, but during deployment, human users rarely restrict themselves to asking questions

tied to a single task. Instead, it is natural for humans to ask compositional questions that combine multiple learned concepts. Figure 5 illustrates such a compositional VQA scenario. Due to the absence of benchmarks targeting compositionality, prior continual VQA methods have primarily focused on mitigating catastrophic forgetting. However, given the open-ended nature of the VQA task, compositional questions, which require multi-step visual reasoning, are both more practical and more reflective of real-world usage. As described in Section 4, we design our continual learning framework to solve such a practical application. To further test our method on human-annotated compositional questions, we create a benchmark, which in this paper we refer to as compositional VQA.

**Benchmark Design.** To create a composed test set, we selected all images from the VQACL test tasks. In total, the ten VQACL tasks share 5,000 images from MSCOCO. We divided these 5,000 images into five groups of students and collected a small subset of composed questions. During the annotation process, we also recorded the task identities to which each composed question belongs. Annotating task identities is important because our goal is to evaluate continual learners only on compositions of previously learned tasks. Table 4 for dataset statistics. Figure 6 shows our sample annotation form. We will make our test subset publicly available.

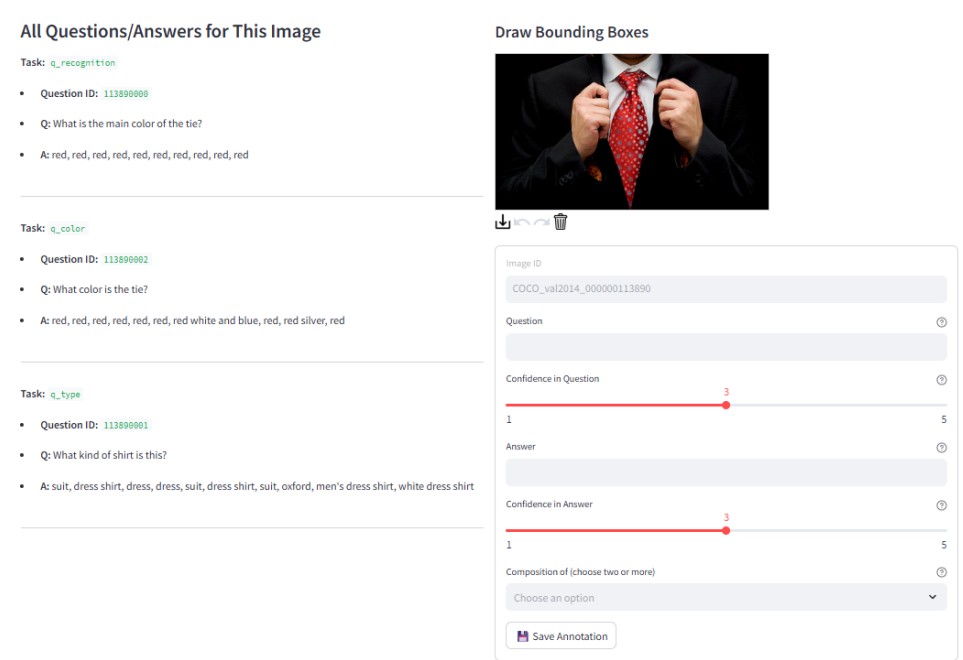

Figure 6: Sample Annotation Form. During the annotation process, annotators were presented with the corresponding images along with all human-asked questions associated with those images. This context was provided to help annotators reason more effectively when creating compositional questions.

# D ADDITIONAL IMPLEMENTATION DETAILS

For our implementation, we adopt LoRA with a scalar of 0.1, dropout set to 0.1, and a bottleneck dimension of 64, without applying layer normalization. The base LLM is Flan-T5-XL, configured with 5 beams, a length penalty of –1, deterministic decoding (nucleus sampling disabled), and a temperature of 1. The Q-Former is initialized with 32 queries, while the vision encoder employs the EVA-CLIP-G ViT model with an image resolution of 224 and FP16 precision. All parameters are frozen except the adapters within the Q-Former. Training is conducted using the AdamW optimizer ($\beta$ ranging from 0.9 to 0.999) with a learning rate of 1e-5, weight decay of 0.05, a layer-wise learning rate decay factor of 1, and 1000 warmup steps.

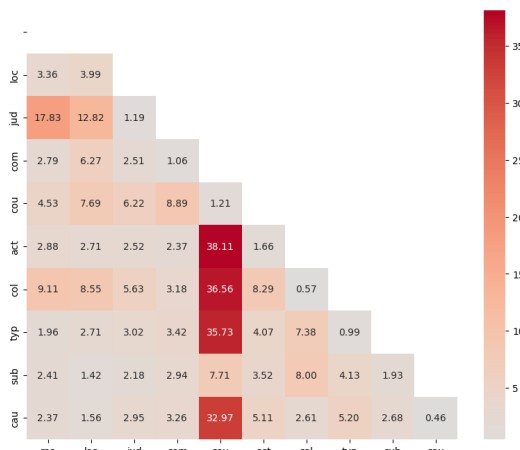

Figure 7: Past vs Self Improvement.

## E    ADDITIONAL RESULTS

**Past vs Self Improvement.** We conduct experiments to evaluate the effectiveness of our method in improving both inter-task and intra-task performance. Specifically, we select each expert individually and measure their performance on the tasks, with and without the knowledge fusion mechanism. Figure 7 presents the performance improvements relative to the setting without inter-expert knowledge fusion. The diagonal entries denote self-improvement. Overall, the heatmap illustrates how our method bridges the knowledge gap among experts.

**Task Dependency.** To further analyze Figure 7 and provide additional explanation, we examine the impact of learning each task on the performance of other tasks. Figure 8 illustrates how knowledge transfer occurs across tasks. We observe that recognition and subcategory exert the strongest influence on other tasks, whereas count and color exhibit the weakest influence. To quantify these dependencies, we employ VL-T5 with a randomly initialized language model backbone.

### E.1    WORKSTATION SETUP

The experiments were conducted on a workstation equipped with three NVIDIA A100 PCIe GPUs (40 GB each), providing a total of 120 GB GPU memory.

## F    LIMITATIONS

Our method addresses both catastrophic forgetting and compositional knowledge in the continual visual question answering setting. The current design focuses on compositions of previously learned tasks, which aligns well with the continual learning paradigm. However, handling future compositional questions (i.e., zero-shot compositions) remains a limitation and lies outside the scope of this work.

## G    SOURCE CODE

We provide a link to the anonymous source code in the ICLR 2026 OpenReview discussion forum as a comment.

## H    USAGE OF LARGE LANGUAGE MODELS

We use Large Language Models (LLMs) solely for polishing writing and correcting grammar.

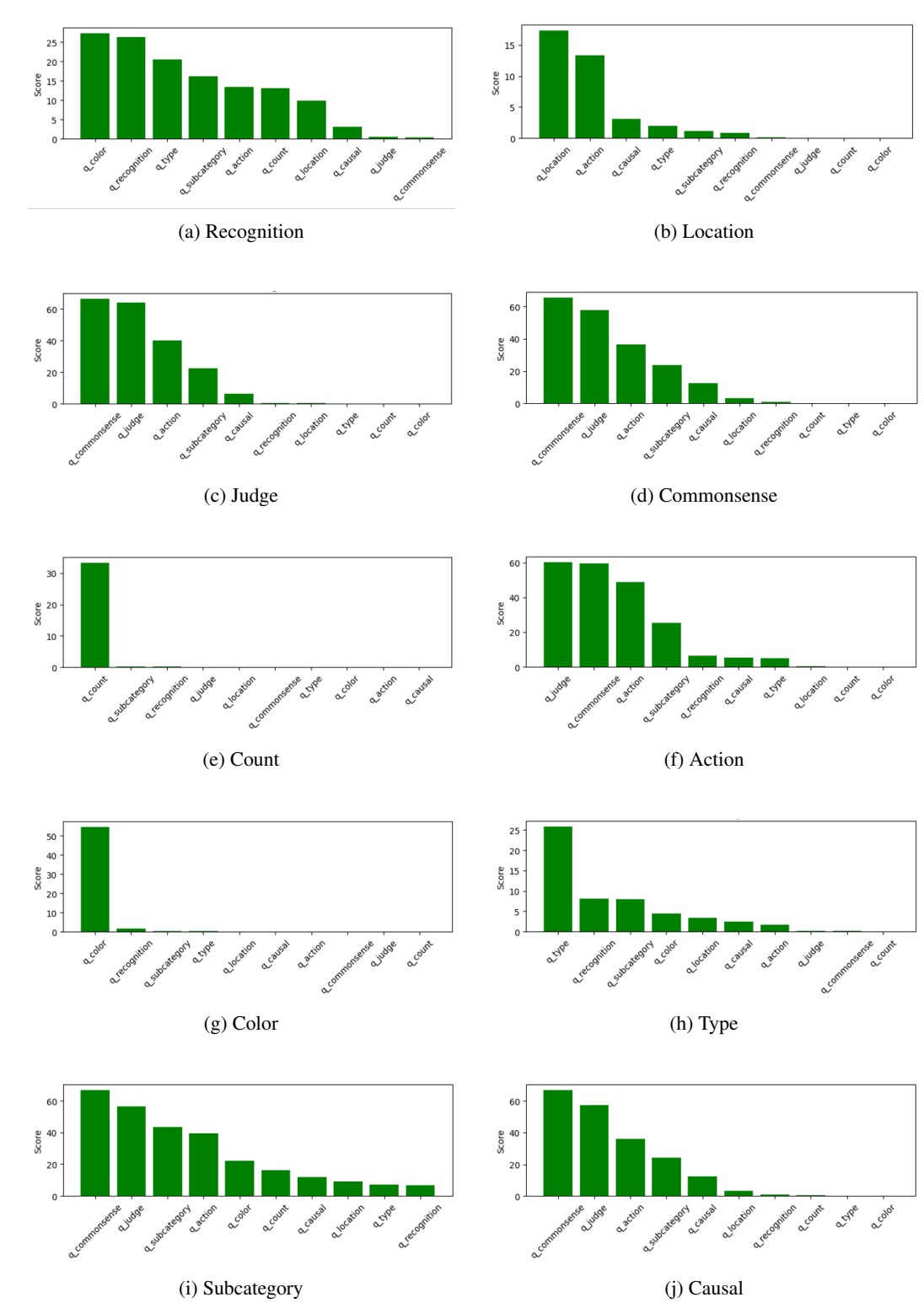

Figure 8: Analysis of impact of learning one task to another.

