# OpenReview forum: "Learning to Compose: Continual Visual QA through a Dual-Purpose Mixture-of-Experts Framework"
_ICLR.cc/2026/Conference — ICLR 2026 Conference Withdrawn Submission_

### Official Review · Reviewer_zpqU · 2025-10-25

**Soundness:** 2
**Presentation:** 2
**Contribution:** 2
**Rating:** 2
**Confidence:** 4

**Summary:**

The paper proposes a dual-purpose Mixture-of-Experts (MoE) framework for continual VQA learning over sequential tasks.  During training, the t-th task receives (t-1) frozen experts and one tunable expert trained jointly on VQA (question answering) and VQG (question generation). With knowledge replay from the (question, image, answer) triples generated from past VQG experts,  this method alleviates forgetting while improving compositional generalization across tasks.

**Strengths:**

- The paper investigates the compositional performance of the model.

**Weaknesses:**

- The novelty is limited. Using a generative model for image or question generation as replay samples for continual learning has been widely used in previous studies [1,2].

[1] Shin, Hanul, Jung Kwon Lee, Jaehong Kim, and Jiwon Kim. "Continual learning with deep generative replay." Advances in neural information processing systems 30 (2017).

[2] Thandiackal, Kevin, Tiziano Portenier, Andrea Giovannini, Maria Gabrani, and Orcun Goksel. "Generative feature-driven image replay for continual learning." Image and Vision Computing 150 (2024): 105187.
- In the paper, Knowledge Fusion refers to replay training on the t-th expert with previously generated (question, image, answer) triplets, while keeping the first (t−1) experts frozen. When the triplet set grows sufficiently large, depending exclusively on the last expert may still result in catastrophic forgetting.
- As shown in Table 1, the performance of the same method such as ER, DER and VQACL varies with different base models (e.g., BLIP2 yields higher results than LLaVA-7B). Whereas, the paper merely reports the results of the proposed method on BLIP2, making the effectiveness not convincing.
- As described in (Huai et al., 2025a), CL-MoE got 51.34 in AP. Whereas, the result in this paper is 44.34. There is a large gap between the results. Concerning the results of other baselines are the same as shown in (Huai et al., 2025a) except CL-MoE, it is quite confusing and not reliable.
- In Line 184-186, it is stated that “Unlike prior approaches that combine parameter isolation and parameter sharing, the latter often reintroduces forgetting due to a shared parameter bottleneck”. Is there an evidence? This statement should be further verified via preliminary studies.
- Some expressions and notations in the paper are presented in an abrupt manner, making it difficult to follow the technical logic. For instance, in line 296, the sentence “Em used on vt ∈ Dt to generate (q̃m, ãm) where m ranges from 1 to t−1” is not clearly defined or explained.
- The description in line 415 could be confusing. It is recommended to explicitly state the accuracy gap compared to the second-best method (56.08%-48.22%=7.86%).

**Questions:**

Please see the weaknesses.

---

### Official Review · Reviewer_3PUY · 2025-11-01

**Soundness:** 2
**Presentation:** 3
**Contribution:** 2
**Rating:** 2
**Confidence:** 3

**Summary:**

This paper tackles continual VQA with an emphasis on compositional reasoning—a practical but underexplored problem. The proposed dual-purpose MoE framework trains experts jointly on VQA and VQG to enable knowledge fusion without replay buffers. While the empirical results are strong and the compositional focus is refreshing, the work suffers from incremental novelty, hand-wavy explanations of key mechanisms, and several puzzling experimental choices that undermine the contributions.

**Strengths:**

- The idea is well-motivated
- The empirical results are strong and the compositional focus is refreshing

**Weaknesses:**

- Compositional benchmark is too small. 200 samples is tiny for drawing strong conclusions.
- The paper essentially combines LoRA + MoE + VQG without introducing fundamentally new ideas. Why does generating questions help with compositional reasoning?
- How exactly does answering generated questions from past experts lead to compositional ability?
- Why not use official numbers of CL-MoE? Reproducing baseline results yourself raises fairness questions.
- Why is your upper bound so much higher than joint training?

**Questions:**

Please answer the questions in the weaknesses section

---

### Official Review · Reviewer_wQjY · 2025-11-01

**Soundness:** 2
**Presentation:** 3
**Contribution:** 2
**Rating:** 4
**Confidence:** 4

**Summary:**

This paper focuses on Continual Visual Question Answering, where a model must sequentially learn multiple visual question answering tasks without forgetting previously learned tasks while maintaining compositional reasoning ability. The paper proposes a framework called Dual-Purpose Experts within a Mixture of Experts. Specifically, the method creates independent dual-purpose experts for each task within a mixture of experts framework, where each expert can perform both VQA and VQG (Visual Question Generation). When learning a new task, previous experts generate pseudo questions and answers in the style of old tasks, and these pseudo samples are used together with current task data to train the new expert, thus enabling knowledge transfer without storing old data. Meanwhile, a language router automatically selects appropriate experts based on the semantic content of input questions. This allows the model to both preserve old knowledge and combine knowledge from different tasks to answer compositional questions. The method is evaluated on three datasets and shows better performance than baselines with almost no forgetting.

**Strengths:**

- I believe this paper addresses an important problem. The authors recognize the compositional reasoning problem that has been overlooked in existing continual learning VQA research and point out that in real-world scenarios, task boundaries are soft and users naturally ask questions that require combining knowledge across tasks, which indeed reflects practical application scenarios.
- The proposed method demonstrates innovation in some aspects, such as unifying VQA and VQG within the same expert and using VQG as an auxiliary task to facilitate knowledge fusion.
- The paper creates a new evaluation benchmark for the experimental setting, which will be valuable for future research.
- Compared to baseline methods, the proposed approach shows strong empirical performance.

**Weaknesses:**

- One of my main concerns is a fundamental conceptual contradiction. The paper claims to solve "continual learning," but it actually avoids continual learning. True continual learning integrates new and old knowledge within the same parameter space, whereas this work creates isolated parameter spaces for each task through physical separation. Therefore, I believe the core contribution should be categorized as multi-task learning with modular design. The paper appears to blur the boundary between "continual learning" and "multi-task learning."
- The conceptual innovation of the proposed method is limited. Essentially, the method trains a LoRA adapter for each task and manages them with MoE. This approach is already quite common in the parameter-efficient fine-tuning field and has even been integrated into some standard libraries.
- The paper emphasizes "without replay buffer," but in reality, having old experts generate pseudo question-answer pairs is a form of conceptual replay. Only the storage format has changed (from data to generator); the essence remains the same and may even be worse due to unstable generation quality. The quality of pseudo data depends on the generation capability of old experts and may introduce errors or biases. Furthermore, I am not entirely convinced that we need VQG here. The introduction of VQG increases system complexity, but the benefits are unclear. As shown in Table 2, except for the count task, improvements for other tasks are quite limited, so it does not seem crucial.
- The proposed method has potential scalability issues. When the total number of tasks becomes large, maintaining two modules for each task and traversing all experts during inference would directly lead to enormous computational overhead. Moreover, when facing a large number of tasks, the reliability of the router is questionable because router failure could directly lead to selecting the wrong expert and producing incorrect answers. I believe the authors should further address this aspect.
- In the experimental part, I think the authors should at least include a comparison with Joint Training as a baseline, and the experimental results lack statistical significance testing for better comparison. I believe the authors could add more analysis regarding the router and pseudo data quality.

**Questions:**

The proposed method creates isolated parameter spaces for each task and freezes old parameters. How is this fundamentally different from multi-task learning with task-specific modules? What makes this "continual" learning if old and new knowledge never truly integrate in the same parameter space?

---

### Official Review · Reviewer_74sg · 2025-11-04

**Soundness:** 3
**Presentation:** 3
**Contribution:** 2
**Rating:** 4
**Confidence:** 4

**Summary:**

The paper addresses challenges in continual VQA using VLMs. Current approaches are hindered by catastrophic forgetting and struggle with compositional generalization. Previous work primarily focused on mitigating forgetting, often overlooking the need for intertask composition in real-world scenarios. The paper introduces a unified dual-purpose MoE framework designed to reduce forgetting and enhance compositional ability without requiring a replay buffer.  Recognizing that existing benchmarks like VQACL lack compositional questions, the paper created a new human-annotated compositional benchmark. This benchmark uses images from the VQACL test set but features questions that require combining skills from multiple previously learned tasks. Experiments on standard and compositional benchmarks demonstrated the method's effectiveness.

**Strengths:**

1. The paper introduces a MoE framework that effectively balances two often conflicting goals in continual learning: stabilizing past knowledge to prevent catastrophic forgetting and enabling compositional reasoning across tasks.

2.  Through parameter isolation using LoRA, the method effectively preserves task-specific knowledge. It achieved the lowest average forgetting among all tested methods on the VQACL benchmark.

3.  Unlike previous approaches that often fail at intertask composition, this framework employs dual-purpose experts (trained on both VQA and Visual Question Generation).

4. The new benchmark addresses a gap in existing research, specifically designed to evaluate compositional reasoning in realistic, open-ended VQA scenarios.

**Weaknesses:**

1. The current routing mechanism relies exclusively on the "surface form of a question" to determine the appropriate expert. It uses a single-layer LSTM to encode the tokenized question sequence. This assumes that task identity is entirely deducible from linguistic cues. It likely fails in scenarios where visual context is required to disambiguate the task (e.g., the question "What is it?" could be an Action, Recognition, or Subcategory task depending on the image content). Relying solely on text may lead to incorrect expert selection when linguistic patterns overlap across tasks.

2. While introducing a human-annotated benchmark is a valuable contribution, its current scale is insufficient for definitive claims. The "Compositional VQA" benchmark consists of only 200 questions. This sample size is too small to ensure statistical significance or to adequately cover the diverse combinations of the ten learned tasks. A few outliers or specific biases in these 200 samples could heavily skew the results (reported as 54.76% for the proposed method ).

3. The inter-expert knowledge fusion relies heavily on the Visual Question Generation (VQG) component and a simple confidence-based filtering mechanism. If an expert's VQG module drifts or learns shortcut definitions of a task, it might generate high-confidence but semantically poor or irrelevant questions. The current confidence score, based on average log-likelihood, guarantees fluency but not necessarily task-relevance or diversity. Poor generations could poison the student model during knowledge fusion.

**Questions:**

See weaknesses

---

### Note · Authors · 2025-11-13

I have read and agree with the venue's withdrawal policy on behalf of myself and my co-authors.